# Individual Determinants of Rest-Break Behavior in Occupational Settings

**DOI:** 10.3390/healthcare9101330

**Published:** 2021-10-06

**Authors:** Gerhard Blasche, Johannes Wendsche, Theresa Tschulik, Rudolf Schoberberger, Lisbeth Weitensfelder

**Affiliations:** 1Center for Public Health, Department of Environmental Health, Medical University of Vienna, A-1090 Vienna, Austria; theresatschulik@hotmail.com (T.T.); lisbeth.weitensfelder@meduniwien.ac.at (L.W.); 2Federal Institute for Occupational Safety and Health, D-01099 Dresden, Germany; Wendsche.Johannes@baua.bund.de; 3Center for Public Health, Department of Social and Preventive Medicine, Medical University Vienna, A-1090 Vienna, Austria; rudolf.schoberberger@meduniwien.ac.at

**Keywords:** theory of planned behavior, rest breaks, rest-break behavior, attitude, subjective norm, behavioral intention, job control

## Abstract

Aims: Work breaks improve well-being, productivity, and health. The aim of this study was to investigate the individual determinants of rest-break behavior during work using the theory of planned behavior (TPB). Methods: The association between attitude, control, and subjective norm and rest-break intention (i.e., taking rest breaks regularly), and rest-break behavior (average number of rest breaks/workhour) was analyzed with stepwise linear regression in a cross-sectional design. The study participants included 109 clerical employees, and 215 nurses. Results: Attitude and control were positively associated with rest-break intention. Intention and control were positively associated with rest-break behavior. The effect of intention was moderated by occupation, with intention being more weakly associated with rest-break behavior in nurses who had less behavioral control. Conclusions: Job control is the major predictor of rest-break behavior, with attitudes playing a minor role, and social norm playing no role. To increase rest-break behavior, a greater extent of job control is necessary.

## 1. Introduction

Rest breaks, i.e., short respites from work with the aim of recovering from work-related stress and/or restoring perceived energy levels, have beneficial effects on the health and well-being of workers. With regard to the immediate effects of rest breaks on well-being, a well-conducted study on self-initiated microbreaks (e.g., drinking a beverage, engaging in physical activity, talking to someone about nonwork issues) in administrative employees found that subjective fatigue decreased and vitality increased in the hour after the break, compared to a work period without a microbreak [1]. Likewise, experimentally induced breaks during a university lecture reduced fatigue and increased vigor in students for at least twenty minutes after the break, compared to lectures without a break [2]. In addition, rest breaks have been found to improve well-being at the end of the work period. For instance, in a randomized controlled study, surgeons taking several short breaks during a surgery of a three-hour duration had lower levels of subjective fatigue following their work compared to physicians performing surgery without breaks [3]. Driving examiners taking breaks between driving exams showed lower levels of distress and higher levels of vigor at the end of their eight-hour workday then when not taking any breaks [4]. In another study, the frequency of relaxing microbreaks taken by office workers in the afternoon was associated with reduced negative affect at the end of the workday, especially for workers reporting high work demands [5]. Similarly, taking a short break in the afternoon, but not in the morning, was associated with greater work engagement during the day [6]. On the basis of these generally well-conducted studies, we can conclude, with some certainty, that rest breaks are associated with immediate and post-work improvements in the various facets of well-being, especially for individuals engaged in demanding work.

In addition to improving psychological well-being, rest breaks were also found to improve attention in monotonous tasks [7,8], to improve task performance [9], to reduce the risk of work-related injuries [10], to reduce musculoskeletal symptoms, both in computer work [11], and manual labor [12], and to reduce the release of stress hormones during work [3,13]. Therefore, we conclude that rest breaks can be effective in improving attention, performance, and various facets of physical health and well-being on a short-term basis.

In most contemporary work settings, taking rest breaks, at least in theory, is at the individual’s discretion. Thus, it is up to the employee to decide whether to take a rest break or not. However, little is known about the factors determining the rest-break behavior of individuals. Considering the importance of rest breaks for health, well-being, and the performance of employees, knowledge regarding the determinants of rest-break behavior could help support employees, for example, by changing their attitudes towards breaks through information on the benefits of rest breaks, or by providing them more control over work scheduling. A common theory describing the determinants of behaviors in general, and health behaviors in particular, is the theory of planned behavior (TPB) [14,15]. The theory proposes that a behavior, insofar as it is under the individual’s control, is determined by the individual’s intention to perform that behavior. Though not every intended behavior is actually executed, meta-analyses have found that behavioral intentions explain more than 30% of the variance in the related behavior [15]. According to the theory of planned behavior, behavioral intention itself is determined by three factors: the individual’s attitude towards the behavior, the attitude of significant others regarding the individual engaging in the behavior (subjective norm), and the extent to which the individual has control over the behavior (behavioral control).

The theory of planned behavior seems well-suited to predict behavior in the work setting. First, according to the theory, *intention* will only be associated with behavior when the behavior is under the employee’s control, i.e., when the employee can “decide at will to perform or not perform the behavior” [14], p 182. Thus, the extent of association between intention and behavior will reflect the actual control that employees have over rest-break behavior. Second, related to this question, the theory also includes the extent of *behavioral control* as perceived by the individual. This does not only reflect the extent of control as determined by external work-related circumstances, but also the extent to which individual employees perceive themselves as capable of performing the behavior [14]. Although, at first glance, the taking of rest breaks does not seem like a difficult behavior to perform, rest breaks not only require the cessation of work, but also require engaging in a suitable activity, such as closing one’s eyes to relax, stretching, or leaving the workplace to take a walk. If these behaviors are to be conducted in the work setting, and under the possible surveillance of supervisors and coworkers, these activities will undoubtedly require a certain amount of self-efficacy to realize [16]. Indeed, recovery-related self-efficacy has been found to foster recovery in work and home environments [17,18].

Third, the *subjective norm*, i.e., the social norms perceived in the workplace, are likely to affect one’s behavior in the workplace for several reasons. An obvious reason is that the employee has to comply with explicit rules [19]. These rules may also apply to the timing, extent, and type of rest breaks accepted at the workplace. In addition, employees are likely to respond to social pressures in their rest-break behavior based on the perception of what supervisors or colleagues want them to do. This type of subjective norm is termed an *injunctive* norm [20]. Furthermore, individuals may also react to the observed behavior of colleagues or supervisors with regard to the taking of rest breaks, a second facet of subjective norm termed a *descriptive* norm [20]. However, these two facets of the subjective norm have been found to be conceptually different constructs and, therefore, need to be considered as separate variables [20].

Fourthly, the third determinant of intention, and subsequently behavior, is the *attitude* towards the behavior. The attitude can be both positive and negative and is associated with the outcome the behavior is expected to have [21]. Attitudes include both affective (e.g., the behavior is enjoyable), and instrumental or cognitive (e.g., the behavior is beneficial), components [14]. Outcome expectations and, thus, the attitudes towards the behavior are derived from information about the behavior, as well as from personal experience with the behavior [22]. Thus, it seems plausible that, for the cognitive component of attitudes, the type and scope of knowledge would affect behavior, but this is not necessarily the case. The association between knowledge and outcome behavior might vary between specific contexts. For example, in the context of nutrition, the association between knowledge and the dietary intake of healthy food is often positive, albeit rather weak [23]. Nonetheless, individual studies also find effects of knowledge on eating behavior [24]. In oral hygiene, knowledge does not always correlate with practice [25]. Generally, however, at least an indirect pathway linking knowledge to health behavior seems to exist, as a study on physical activity shows [26]. As far as we know, no study has yet investigated the association between knowledge or attitudes and the taking of rest breaks.

The present study sought to investigate the impact of the four TPB components: intention, attitude, subjective norm, and behavioral control on rest-break behavior in a sample of working individuals in an effort to highlight factors promoting and/or discouraging rest-break behavior. We decided to study these associations in a combined sample of nurses and clerical (i.e. administrative) employees, as these two occupational groups differ with respect to several key job characteristics, such as occupational stress, job control, and shift length. Statistically, this should increase the variance of the TPB variables and, thus, the reliability of the model while, at the same time, increase the generalizability of the results. Compared to clerical employees, nurses generally have higher levels of stress [27,28,29]. In addition, nurses and other healthcare employees experience a high level of disruptive interruptions, thus limiting control over their work [27]. It has been reported that, especially among nurses, rest breaks are often missed, interrupted, or delayed [30]. Smoking also plays an important role; nonsmoking nurses are nearly twice as likely to miss breaks as their smoking colleagues [31]. In addition, working time is different for the two occupations, as nurses often have twelve-hour shifts, compared to the eight-hour days of clerical employees. According to the applicable Austrian Working Times Act (”Arbeitszeitgesetz”), rest breaks of thirty minutes in total are mandatory for both occupational groups on days with more than six hours total work time. In addition, supplemental short rest breaks (<15 min) are typical for both occupational settings [1,9,30].

According to the TPB, we assume that a higher (positive) attitude, a higher subjective norm, and a higher behavioral control independently predict higher rest-break intention (Hypothesis 1), and better rest-break behavior (i.e., higher rest-break frequency; Hypothesis 2). In addition, and in line with the TPB [32], we assume that rest-break intention will explain (i.e., mediate or indirectly affect) the positive relationship between attitude and the subjective norm, as well as, in part, behavioral control on the one hand, and rest-break behavior on the other (Hypothesis 3). Because of the presumably greater extent of job control that clerical employees have compared to nurses [30], we expect the positive association between the intention and the actual behavior, i.e., the rest-break frequency, to be stronger for clerical employees than for nurses (Hypothesis 4). The aim of the present study is to highlight the relevant individual factors affecting the rest-break behavior of employees. This should help to devise evidence-based strategies promoting the taking of rest-breaks that could be beneficial both for employees, and for those administrating programs aimed at reducing occupational stress.

## 2. Material & Methods

### 2.1. Study Design and Sample

The study utilized a cross-sectional design and a convenient sample of nurses and clerical employees of the General Hospital of Vienna. The administrative staff of the General Hospital of Vienna, as well as the nurses of the various clinics of this hospital, were contacted in February 2016 via email by the heads of the corresponding departments (head nurse, head of the human resource department), who forwarded an invitation letter by the third author of this paper asking them to participate in an online survey (via www.soscisurvey.de, accessed on 30 March 2016) on rest breaks during worktime. A link to the online survey was provided, together with a password to access the survey. The inclusion criteria were a minimum of twenty workhours per week, and a minimum of six workhours per day, as employees are required to take breaks if they work six hours or more per day according to the Austrian Working Times Act. Participation in the survey was voluntary and anonymous, meaning that respondents could not be identified, and managers did not receive feedback on employee participation or responses. A total of 365 individuals initially responded to the questionnaire. After excluding employees with incomplete data (*n* = 38), and limited daily (<6 h/day, *n* = 2) and weekly (<20 h/week, *n* = 1) working hours, both restricting the right or the opportunity for work breaks, we used the remaining sample of 324 employees (66 men, 258 women, mean age = 41.4, SD 11.2, age range 18–64) for further analyses. Sample characteristics are presented in Table 1.

### 2.2. Study Variables

In the survey, we assessed the rest-break behavior of the employees by asking them if they take lunch breaks, and how many rest breaks they take additionally during a typical workday. To determine the dependent variable rest-break frequency, i.e., the relative number of rest breaks taken per hour, the sum of the lunch breaks and the additional rest breaks was calculated and divided by the number of hours typically worked per day.

The main independent variables were the four determinants of behavior, according to the theory of planned behavior: attitude, subjective norm, behavioral control, and intention. The items were generated and phrased in accordance with the description of questionnaire construction stated in [33]. Additional items assessing salient beliefs were generated on the basis of nine interviews with nurses, clerical employees, and technicians. In these interviews, the advantages and hindrances of regularly taking rest breaks were explored. To assess the structure of the three TPB variables, an exploratory factor analysis was conducted in advance, revealing three factors. All items, except the one assessing the descriptive norm (“most of my colleagues in comparable positions regularly take breaks from work”), loaded in the expected factors, the latter loading in the control factor (factor loading = 0.60). Therefore, this item was removed. The final TPB variables, plus the factor loadings, are displayed in Table 2. The internal consistency of the scales (Cronbach’s alpha) were satisfactory, with *α* = 0.81 for attitude, *α* = 0.71 for subjective (injunctive) norm, and *α* = 0.75 for behavioral control.

As it is likely that nurses have less job control compared to administrative staff, e.g., less control over the timing of breaks because of patient care obligations [30], we assumed that intention would affect rest-break behavior in nurses to a smaller degree than in clerical employees and, hence, we included an interactive term between occupation and intention (occupation * intention) as an additional variable.

Rest-break intention was assessed with a single item (“I intend to take rest breaks regularly the next three months”). All TPB items were assessed on 7-point Likert scales, ranging from 1 = ”absolutely not”, to 7 = ”totally applies”.

### 2.3. Statistical Analyses

The independence of the three TPB variables, attitude, behavioral control, and subjective norm, was determined by factor analysis with varimax rotation and a factor extraction based on an eigenvalue > 1. Group differences were calculated using one-way variance analysis, or a chi-squared test with Bonferroni-Holm adjustments for multiple testing. The association between the independent and dependent variables was analyzed using hierarchical linear regression analyses. Overall, two analyses were calculated. The first analysis included rest-break intention as the dependent variable, and work-related variables (occupation, average working hours/day), demographic variables (sex, age, smoking status), and the variables according to the TPB (attitude, subjective norm, control), as the independent variables. The second analysis included rest-break frequency (total breaks/hour) as the dependent variable, and the same independent variables as described above, with the addition of rest-break intention, as independent variables. Thereby, separate models were calculated, i.e., groups of variables and/or individual variables were entered in the analysis successively in order to be able to observe the amount of variance in the previously included variables explained by the variables included at a later step. In order to test the indirect effects of rest-break intention on the relationship between the TPB variables and rest-break frequency (Hypothesis 3), we used the PROCESS Plugin from Hayes [34]. To test the proposed interaction effect between the occupational groups and rest-break intention (Hypothesis 4), we z-standardized all variables and calculated the respective interaction term. The level of significance was set to 5%. All calculations were conducted using standard software (SPSS 26, IBM corp.).

## 3. Results

The characteristics of the two study groups (nurses, clerical employees), as well as the means and standard deviations of the study variables, are displayed in Table 1. Compared to clerical employees, nurses smoked less, had longer workdays due to twelve-hour shifts, took lunch breaks less often, and took fewer rest breaks per working hour. In addition, nurses tended to have a more positive attitude towards rest breaks but experienced less control over the taking of rest breaks. The results of the factor analysis, including the items of the TPB variables, are shown in Table 2. The analysis supports the notion of three independent factors with an Eigenvalue of ≥ 1.15, items loading highly in the expected factors.

The results of the linear regression analyses are displayed in Table 3. The predictors of rest-break intention, attitude, and behavioral control, but not the injunctive form of the subjective norm, were positively associated with intention, thus partly supporting Hypothesis 1. Rest-break intention was not associated with any of the individual characteristics.

Regarding rest-break frequency as a behavioral outcome, i.e., the average number of rest breaks individuals took per hour, attitude, and behavioral control, but not the injunctive form of the subjective norm, positively predicted rest-break behavior (Model 2), thus partly supporting Hypothesis 2.

Using the PROCESS Plugin for SPSS by Andrew Hayes [34] (bias-corrected bootstrapping with 5000 samples; adjusted for other variables shown in Table 3), we found that rest-break intention indirectly affected the relationships between positive attitudes (*b* = 0.05, *SE* = 0.02, 95% CI [0.016, 0.092]), as well as behavioral control (*b* = 0.08, *SE* = 0.03, 95% CI [0.025, 0.130]) and rest-break frequency, but not between the subjective norm and rest-break frequency (*b* = 0.02, *SE* = 0.01, 95% CI [−0.005, 0.043]). This partly supports Hypothesis 3. Together, the four variables of the theory of planned behavior explained 11% of the variance in rest-break frequency.

Finally, the association between rest-break intention and rest-break frequency was stronger for clerical employees (*b* = 0.05, *t* = 4.47, *p* < 0.001) than for nurses (*b* = 0.01, *t* = 2.31, *p* = 0.022), as can be seen in the significant interactive term (Table 3, Model 4), thus supporting Hypothesis 4. This relationship is illustrated in Figure 1 (slope tests and visualization of the interaction effect with Excel template from Jeremy Dawson [35]). Rest-break intention affected behavior to a greater degree in clerical employees than in nurses.

Apart from the TPB variables, rest-break frequency was also strongly associated with the sex, age, smoking status, and occupation of the individuals. A higher rest-break frequency was reported by males compared to females, by younger compared to older employees, by smokers compared to nonsmokers, and by clerical employees compared to nurses (Table 3).

Plot based on linear regression analysis, as illustrated in Table 3.

## 4. Discussion

The present study sought to uncover the individual determinants of rest-break behavior in two occupational groups differing markedly in various work characteristics, such as shift length and work pressure: nurses and clerical employees. For this purpose, the theory of planned behavior (TPB) was applied, which seems well-suited to predict behavior in the occupational setting, as argued above. The TPB assumes that attitudes, the subjective norm, and behavioral control determine intention and, subsequently, behavior. The results indicate that higher control and higher intention were predictive of taking rest breaks more frequently. The intention to take rest breaks, furthermore, was associated with a more positive attitude towards rest breaks and stronger behavioral control. However, the association between intention and rest-break behavior was small, whereas individual characteristics, including the smoking status and the type of occupation, explained break behavior to a greater degree.

*Attitude*, generally, can be considered a relevant predictor of the intention to initiate a behavior and, in turn, to perform the behavior, as attitudes can be modified by providing information regarding the consequences of a behavior [36,37]. However, in the present rest-break study, the relationship between the behavioral attitude and the intention was smaller than that found in meta-analyses focusing on other health variables (*r* = 0.30 versus *r* = 0.51) [32], although attitude did predict both rest-break intention and rest-break behavior to some degree. There are two main explanations for the limited impact of intention in the present study. First, beliefs as assessed in the present study were probably not sufficiently salient for the occupational groups under investigation because the emphasis of the study was the maintenance of well-being and performance [38]. Thus, future studies should investigate other causes for taking rest breaks, such as drinking, eating, or smoking, in more detail [39]. Second, employee knowledge of the health-related benefits of rest breaks generally may be insufficient, although differences between the two occupations were apparent. Nurses had a somewhat stronger positive attitude towards rest breaks than administrative employees, which might be a consequence of their greater health-related knowledge. Moreover, the difficulty of taking breaks for nurses may affect their rest-break attitude by increasing desirability [40]. Future studies will have to address these issues by assessing the outcome expectancies of rest breaks more broadly, and by studying the impact of health-related knowledge.

We assumed that the *subjective norm* would be a relevant direct or indirect predictor of rest-break behavior, considering that employees have to consent to explicit and implicit rules at the workplace [19]. However, this was not the case. The subjective norm was neither associated with rest-break intention, nor with rest-break behavior itself. It should be noted that the subjective norm assessed in the present study was based only on the *injunctive norm* (i.e., belief as to what supervisors or colleagues want one to do) because the item assessing the descriptive norm (i.e., the rest-break behavior of colleagues) was conceptually associated with control and, thus, omitted. This fact may have reduced the predictive value because some studies found that the injunctive norm was less strongly related to intention and to behavior compared to the descriptive norm, especially for socially undesirable behaviors [20,38]. An explanation for the failure of the injunctive norm to affect intention, as well as behavior, may be that the taking of rest breaks as a means to promote well-being and performance are hardly an issue at the workplace, so that even if employees believed supervisors or colleagues wanted them to take breaks, they still may be hesitant to actually take them. Thus, future studies would have to look into the actual discourse regarding rest breaks in the workplace.

Although the effects of the *descriptive norm* were not analyzed as stated above, it is likely that this facet of the subjective norm is associated with rest-break intention and behavior, keeping in mind its close association with behavioral control, and the strong association between control and behavior as discussed below. However, future studies will have to explicitly verify this assumption.

*Behavioral control* was one of the strongest predictors of both rest-break intention and behavior, and it affected break behavior both via direct (association with behavioral control) as well as an indirect (differences in the impact of intention between occupations) pathways. This indicates that taking rest breaks depends on the extent of control an employee has over his or her work, which is in line with a recent study on the factors affecting rest-break behavior in nurses in Germany [41]. Behavioral control encompasses both external control related to job characteristics, as well as internal perceived control [14]. The finding that nurses had less control over the taking of rest breaks than clerical employees indicates that rest-break control is related to job characteristics. The latter conclusion is in line with the smaller association between intention and behavior found for nurses compared to clerical employees, since intention can only determine behavior if the behavior is under the individual’s control [14].

*Intention* a key variable of the TPB, predicted the taking of rest breaks. This indicates that rest-break behavior is at least, to some extent, a voluntary and goal-directed behavior, which is in line with the results from another study on the individual determinants of rest breaks, despite differences in the measures of intention [42]. However, the present association was smaller than generally found (*r* = 0.30 versus *r* = 0.48) [32], and intention only explained approximately 10% of the variance in taking rest breaks. There are several potential reasons for this small association. First, the overall intention to take rest breaks might not have been sufficiently strong to significantly determine the behavior [14], indicating that employees may not explicitly plan their rest breaks in advance but, rather, take them spontaneously if an opportunity arises. Another explanation, related to research on other health behaviors, is that initiating and taking rest breaks may be less a volitional behavior than a habit, automatically triggered by environmental stimuli [43]. Moreover, rest-break behavior can not only be described by the frequency of breaks (as assessed in this study), but also by the total break duration [9,30]. Future studies need to consider both behaviors in combination since, for some workers, the intention to take breaks regularly might also mean taking a long lunch break every day. And finally, as discussed above, limited job control might impede rest-break behavior, for example, by interruptions or time pressures, even if it had been intended [14]. Future studies will have to clarify these questions, among others, by including measures for the habitual taking of rest breaks.

Our results further suggest that factors not related to the TPB affect rest-break behavior as well. In the present study, this was smoking status, occupation, sex, and age. Indeed, the present study confirmed previous results that showed that smoking employees take more breaks than their nonsmoking counterparts [30,31]. Undoubtedly, smokers are more inclined to take breaks because of their addiction [44], but social factors may also play a role, such as escaping everyday routine [45]. Although smoking, traditionally, has been considered an eligible reason for work breaks, it would be desirable for employees to allow frequent breaks for all employees, and to motivate and support nonsmokers in taking such additional breaks as well.

Compared to the TPB variables, smoking and the other demographic factors explained break behavior to a larger degree, thus corroborating the limitations of the TPB [38]. In a recent publication, Ajzen noted the potential of enriching the TPB by including background factors and variables shaping habit formation [46]. Moreover, the Health Action Process Approach model from Schwarzer [47] tries to overcome the limitations of the TBP by describing the volitional processes of behavior formation (i.e., planning, initiation, maintenance, and relapse management) in more detail, and by adding additional factors, such as situational barriers and resources, different types of self-efficacy, and self-monitoring. Beyond this social and health psychology perspective, however, such factors have not yet been systematically studied in the rest-break literature [30]. Therefore, the integration of these variables into theoretical models is desirable in order to stimulate further research. For instance, a more comprehensive model predicting rest breaks should include situational factors with nudging properties, such as task completion or a friendly interruption by a coworker [48], work and organizational factors relating to impaired (e.g., time pressure), or successful (e.g., social support), recovery behavior, and individual motives for work breaks not primarily associated with recovery, such as smoking or chatting [30,49].

## 5. Study Limitations

Our study is not without limitations. First, we conducted a retrospective and cross-sectional questionnaire study. Therefore, we can only interpret associations, not causal relations between variables. Moreover, the subjective estimates of break behavior might have been biased by cognitive factors relating to memory or social desirability [50]. However, the mean and standard deviation of rest-break frequencies in the present study were similar to that of a previous diary study assessing rest breaks over several work days [42], suggesting that the assessment of rest breaks in the present study is valid. Furthermore, our research focused on two specific occupations differing in the amount of control over working time. It remains unclear to what extent our results can be generalized to other occupations with different job characteristics because lower time control is positively related to workload [51], which might further impair break behavior [30,41]. This should be investigated in future research, additionally, by assessing key job characteristics.

## 6. Conclusions

Rest breaks are known to improve the well-being, health, and performance of employees, but factors that affect the triggering of breaks are largely unknown. Our results reveal that the break intention of employees, i.e., the voluntary planning of break behavior, is such a driver, even though the association between intention and rest-break behavior was small. In addition, behavioral control had a positive impact on rest-break behavior, exceeding that of intention, thereby also explaining the differences between the two occupations. Therefore, next to employee education (i.e., by supervisor or occupational safety engineer), which might strengthen positive attitudes towards recovery behavior, the importance of job control as a factor determining rest-break behavior has to be taken into account. Our results suggest that organizations should increase the temporal flexibility of employees (i.e., give them more control regarding the duration and timing of tasks during the workday) if they intend to improve their break frequencies. The limited impact of intention also suggests that making use of situational factors in terms of nudging (e.g., digital break reminders or ‘break’ posters on the walls of the work site) may help foster rest-taking behavior.

## Figures and Tables

**Figure 1 healthcare-09-01330-f001:**
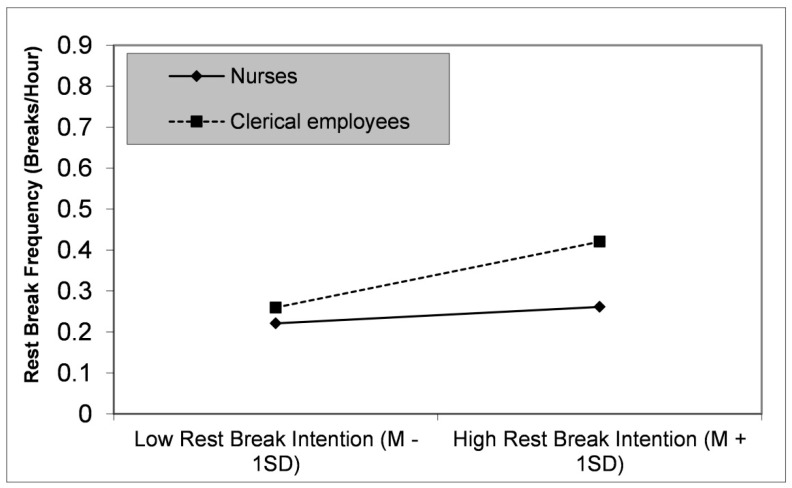
Effect of the interaction between occupation and rest-break intention on rest-break frequency.

**Table 1 healthcare-09-01330-t001:** Descriptive statistics of variables for both occupational groups and the total sample.

		Nurses (*n* = 215)				Clerical Employees (*n* = 109)				Total Sample (*N* = 324)				
Variables	Range	*M*	*SD*	*n*	%	*M*	*SD*	*n*	%	*M*	*SD*	*n*	%	*p*
Age (years)	18–64	41.9	10.8			40.5	11.9			41.4	11.2			0.296
Sex (women)	0–1			179	83.3			79	72.5			258	79.6	0.023
Smoker (yes)	0–1			**52**	**24.2**			**43**	**39.4**			**95**	**29.3**	**0.004**
Working hours/week	20–56	38.6	8.2			38.9	7.1			38.7	7.8			0.763
Working hours/day	6–13	**10.6**	**2.0**			**8.7**	**1.2**			**10.0**	**2.0**			**<0.001**
Lunch break (yes)	0–1			**127**	**59.1**			**87**	**79.8**			**214**	**66.0**	**<0.001**
Total rest breaks/hour	0–0.9	**0.24**	**0.11**			**0.35**	**0.18**			**0.28**	**0.14**			**<0.001**
Attitute	4–28	**22.8**	**4.7**			**21.1**	**5.7**			**22.2**	**5.1**			**0.004**
Subjective (injunctive) norm	2–14	7.7	2.9			8.4	3.0			7.9	3.0			0.046
Behavioral control	3–21	**9.5**	**4.2**			**13.1**	**4.6**			**10.7**	**4.7**			**<0.001**
Rest-break intention	1–7	4.4	1.6			4.6	1.8			4.5	1.7			0.332

Notes. Mean values and standard deviations of frequencies and percentages are displayed; significant values (Bonferroni-Holm adjusted: *p* < 0.004) indicating group differences are printed in bold.

**Table 2 healthcare-09-01330-t002:** Questionnaire items and factor analysis for TPB variables.

		Factor		
		1	2	3
Attitude (positive)	If I took regular (i.e., every 1–2 h) breaks from work, that would be good.	0.82		
	If I took regular (i.e., every 1–2 h) breaks from work, that would be pleasant.	0.82		
	If I take regular breaks from work, it helps my concentration.	0.79		
	Taking regular breaks from work keeps me from getting exhausted/burned out.	0.76		
Behavioral control	I am confident that I will be able to take regular breaks from work in the next three months.		0.84	
	It is up to me to take regular breaks from work over the next three months.		0.78	
	Due to the type or extent of work it is difficult to take breaks from work.		−0.76	
Subjective (injunctive) norm	Most of my colleagues approve of my taking regular breaks from work.			0.87
	Most of my supervisors approve of my taking regular breaks from work.			0.85

Notes. Factor loadings < 0.30 are not shown; total explained variance was 68.8%.

**Table 3 healthcare-09-01330-t003:** Linear regression analyses predicting rest-break intention and rest-break frequency (total breaks per hour).

	Rest-Break Intention		Rest-Break Frequency (Total Breaks Per Hour)			
Model	1	2	1	2	3	4
Sex	−0.04	0.01	−0.13 *	−0.10 *	−0.10 *	−0.10 *
Age	−0.08	−0.06	−0.12 *	−0.12 *	−0.11 *	−0.11 *
Smoking status	0.01	−0.01	0.27 **	0.26 **	0.26 **	0.26 **
Average working hours/day	−0.04	0.01	−0.06	−0.03	−0.03	−0.03
Occupation	0.02	−0.08	0.26 **	0.20 **	0.21 **	0.21 **
Attitude		0.32 **		0.15 **	0.09	0.07
Subjective (injunctive) norm		0.09		0.04	0.02	0.04
Behavioral control		0.47 **		0.27 **	0.19 **	0.15 *
Rest break intention					0.16 **	0.17 **
Occupation * Intention						0.18 **
Δ*R*^2^	0.01	0.31 **	0.23 **	0.09 **	0.02 **	0.03 **
*R*²	0.01	0.32 **	0.23 **	0.31 **	0.33 **	0.36 **

Notes. *N* = 324, sex (male < female), smoking (nonsmoker < smoker), and occupation (nurses < clerical). Standardized regression coefficients *b* are shown. * *p* < 0.05, ** *p* < 0.01.

## Data Availability

Upon request, the corresponding author will provide the data for private use.

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
