# Peer review of "Individual Determinants of Rest-Break Behavior in Occupational Settings"

_healthcare, 2021, doi:10.3390/healthcare9101330_

Round 1

Reviewer 1 Report

The article presents the well-written introduction, but at the end, when talking about hypotheses, it is stated in line 130 that there will be a mediation between the attitude and the subjective norm. However, regressions with mediating variables are not calculated, but hierarchical regressions. I think the authors are confused with the term mediation. In a mediation model, the independent variable does not directly influence the dependent variable, but instead does so through a third variable (mediating variable). And in a hierarchical or step-by-step regression you cannot introduce mediating variables. Therefore, it cannot be hypothesized or assumed that rest break intention will mediate the positive relationship between attitude and subjective norm, as it has not been tested.

In the following sections there is quite a bit of confusion. Section 2.1 only describes the sample and procedure, not the study design. It is convenient to indicate when the data was collected, since during and after confinement certain habits have changed with respect to rest breaks. It should also be indicated if the data was collected anonymously, so that the people who responded could not be identified, since it is mentioned that the participants were contacted through the department heads. Table 1 needs to be completely redone. Data for means and frequencies, or standard deviation and percentage cannot be presented in the same column. It is too confusing, and it is also very different data. The table can be divided into two parts and in the first row offer the variables with means and standard deviations, and in the following rows, and conveniently indicated before offering the data, the variables with frequencies and percentages. Also in this table, it seems that the minimum and maximum values are given in certain variables in brackets. This information must be provided separately in the table in two columns (minimum and maximum), and clearly indicate in the header what information is offered: descriptive statistics of the sample. If there are abbreviations, their meaning should be explained at the bottom of the table. However, the Table Footnotes indicate what should appear in the header. On the other hand, a significance level is indicated in the last column and at the bottom of the table it is said that the significant values have been printed in bold. But there is nothing in bold, nor is it known which statistical contrast has been carried out or between which variables or groups. It is neither written in the text nor indicated in the table, and must be said in both places.

In point 2.2 the description of the variables is mixed with the analyzes performed (line 162a 164). Immediately afterwards, the results of an exploratory factor analysis (EFA) and Chronbach's alpha are offered, the results of which should appear in the Results section. These analyzes should first be described in the Analysis section (the EFA is named but it is not said that alpha will be calculated) and their results should appear in the Results section. Line 169 describes that the TPB items were measured with a 7-point Likert-type scale, but it should also be indicated which were the verbal anchors of the extremes: Strongly disagree and Strongly agree? When describing the independent variables entered in the regression, there is a variable that does not appear in the table: average working hours / day. And another variable appears in the table with another name: work type is occupation in the table. It is convenient that the name is the same to facilitate reading. Furthermore, the table shows what appears to be an interaction between occupation and intention. In section 2.2 no mention is made of such interaction, and if the term is not created first in SPSS and is entered in the analysis as one more independent variable, SPSS does not include it. It would be convenient to indicate the existence of this variable in the text and justify its creation, since there are many variables in the model and no further interactions have been calculated. Or at least they are not featured in the table.

In the Results section, line 202 shows the mediation effect that the authors do not verify. The intention of rest is not a mediating or moderating variable. It is the dependent variable in the regression. On line 213, the authors refer to Table 1, but this is probably Table 3. Several regression models are listed in this table. If the authors have calculated a stepwise regression, they can directly provide the latest model with the independent variables entered as relevant in the regression. The inclusion of the different models can lead to confusion, and in the end the important thing is to know which are the relevant variables in the model and the percentage of variance that they explain.

It would also be a good idea to write the names of the dependent variables in the header of Table 3 as they appear in the table, because otherwise it will lead to confusion at first. It is indicated in the header rest break frequency, while in the table it appears Total rest breaks / hour. Although it is explained in the text, the table must be completely clear. You can write the header by including in parentheses the way that rest break frequency is described in the table.

On the other hand, the authors should clarify the meaning of the variable occupation*intention. It seems like an interaction that is not described anywhere, and that is not mentioned in the discussion either. Likewise, the profile of the type of person who seems to have a higher prediction of rest break frequency is not mentioned: it would be a male, younger than older, smoker, clerical employee with high behavioral control, rest break intention and occupation*intention. And again, we still don't know the meaning of that interaction. I suspect that, since intention is measured with a Likert-type scale item and the reference group in occupation is clerical, this interaction indicates that clerical employees with the highest intention are those who predict the most rest break frequency. But this information has to be explained and the variable clearly described, as well as its meaning in the interaction, by the authors.

Author Response

Reviewer 1

Comment 1

The article presents the well-written introduction, but at the end, when talking about hypotheses, it is stated in line 130 that there will be a mediation between the attitude and the subjective norm. However, regressions with mediating variables are not calculated, but hierarchical regressions. I think the authors are confused with the term mediation. In a mediation model, the independent variable does not directly influence the dependent variable, but instead does so through a third variable (mediating variable). And in a hierarchical or step-by-step regression you cannot introduce mediating variables. Therefore, it cannot be hypothesized or assumed that rest break intention will mediate the positive relationship between attitude and subjective norm, as it has not been tested.

We thank the reviewer for this point. To clearly demonstrate the mediation effects, we calculated these using the PROCESS Plugin for SPSS by Andrew Hayes. This information was added to the section materials & methods/statistical analyses (line 9) as well as to the result section (4th paragraph). The proposed mediating effects of rest break intention were generally confirmed.

“Using the PROCESS Plugin for SPSS by Andrew Hayes (https://www.processmacro.org/index.html; bias-corrected bootstrapping with 5000 samples; adjusted for other variables shown Table 3), we found that rest break intention indirectly affected the relationships between positive attitudes (b = .05, SE = .02, 95% CI [.016, .092]) as well as behavioral control (b = .08, SE = .03, 95% CI [.025, .130]) and rest break frequency but not between subjective norm and rest break frequency (b = .02, SE = .01, 95% CI [-.005, .043]). This supports Hypothesis 3 partially.”

Comment 2

In the following sections there is quite a bit of confusion. Section 2.1 only describes the sample and procedure, not the study design. It is convenient to indicate when the data was collected, since during and after confinement certain habits have changed with respect to rest breaks. It should also be indicated if the data was collected anonymously, so that the people who responded could not be identified, since it is mentioned that the participants were contacted through the department heads.

We thank the reviewer for pointing out these omissions. We amended section 2.1. by adding information on the study design, assessment time and the anonymity of data assessment: “The participation in the survey was voluntary and anonymous, meaning that managers did not receive feedback on employee responses.” Moreover, we report that employees were surveyed in 2016.

Comment 3

Table 1 needs to be completely redone.

Data for means and frequencies, or standard deviation and percentage cannot be presented in the same column. It is too confusing, and it is also very different data. The table can be divided into two parts and in the first row offer the variables with means and standard deviations, and in the following rows, and conveniently indicated before offering the data, the variables with frequencies and percentages.

Also in this table, it seems that the minimum and maximum values are given in certain variables in brackets. This information must be provided separately in the table in two columns (minimum and maximum), and clearly indicate in the header what information is offered: descriptive statistics of the sample. If there are abbreviations, their meaning should be explained at the bottom of the table. However, the Table Footnotes indicate what should appear in the header. On the other hand, a significance level is indicated in the last column and at the bottom of the table it is said that the significant values have been printed in bold. But there is nothing in bold, nor is it known which statistical contrast has been carried out or between which variables or groups. It is neither written in the text nor indicated in the table, and must be said in both places.

Thank you very much for pointing out these issues. We have revised the table accordingly and changed the header to “Descriptive statistics of variables for both occupational groups and the total sample.”

Comment 4

In point 2.2 the description of the variables is mixed with the analyzes performed (line 162a 164). Immediately afterwards, the results of an exploratory factor analysis (EFA) and Chronbach's alpha are offered, the results of which should appear in the Results section. These analyzes should first be described in the Analysis section (the EFA is named but it is not said that alpha will be calculated) and their results should appear in the Results section.

We thank the reviewer for pointing out these issues. We described the use of the final factor analysis in section 2.3. (Statistical Analyses) as suggested. The result of this factor analysis is referred to in the results section (first paragraph, line 6). However, we consider Cronbach Alpha to be an indicator of the quality of the variable (internal consistency), an information which in our opinion should be stated in the variable section. 

Comment 5

Line 169 describes that the TPB items were measured with a 7-point Likert-type scale, but it should also be indicated which were the verbal anchors of the extremes: Strongly disagree and Strongly agree?

Thank you for pointing out this omission, we now provide the verbal scale anchors (section 2.2., last sentence). “All TPB items were assessed on 7-point Likert scales, ranging from 1=”absolutely not” to 7=”totally applies”.”

Comment 6

When describing the independent variables entered in the regression, there is a variable that does not appear in the table: average working hours / day.

And another variable appears in the table with another name: work type is occupation in the table. It is convenient that the name is the same to facilitate reading.

Furthermore, the table shows what appears to be an interaction between occupation and intention. In section 2.2 no mention is made of such interaction, and if the term is not created first in SPSS and is entered in the analysis as one more independent variable, SPSS does not include it. It would be convenient to indicate the existence of this variable in the text and justify its creation, since there are many variables in the model and no further interactions have been calculated. Or at least they are not featured in the table.

Thank you for pointing out these inconsistencies. We added the variable average working hours / day to Table 3. Secondly, we used “occupation” both in the variable section as well as in Table 3. Thirdly, we mentioned the interactive variable and justified its use both in section 2.2. (“Due to the fact that nurses have less job control [30], we assumed that intention would affect rest break behavior in nurses to a smaller degree than in clerical employees and hence included an interactive term between occupation and intention (occupation * intention) as additional variable.”).

Comment 7

In the Results section, line 202 shows the mediation effect that the authors do not verify. The intention of rest is not a mediating or moderating variable. It is the dependent variable in the regression.

On line 213, the authors refer to Table 1, but this is probably Table 3. Several regression models are listed in this table. If the authors have calculated a stepwise regression, they can directly provide the latest model with the independent variables entered as relevant in the regression. The inclusion of the different models can lead to confusion, and in the end the important thing is to know which are the relevant variables in the model and the percentage of variance that they explain.

As stated above (Comment 1), we directly calculated the mediating effects of rest break intention using the PROCESS Plugin for SPSS and state the results of this analysis in the result section (4th paragraph): “Using the PROCESS Plugin for SPSS by Andrew Hayes (https://www.processmacro.org/index.html; bias-corrected bootstrapping with 5000 samples; adjusted for other variables shown Table 3), we found that rest break intention indirectly affected the relationships between positive attitudes (b = .05, SE = .02, 95% CI [.016, .092]) as well as behavioral control (b = .08, SE = .03, 95% CI [.025, .130]) and rest break frequency but not between subjective norm and rest break frequency (b = .02, SE = .01, 95% CI [-.005, .043]). The four variables of the theory of planned behavior together explained 11% of the variance in rest break frequency.”

The statement in line 213 does indeed refer to Table 3, we corrected this mishap and explicitly described the relationship between rest break behavior and individual characteristics (section 3, last paragraph): ” Apart from the TPB variables, rest break frequency was also strongly associated with the individuals’ sex, age, smoking status and occupation. A higher rest break frequency was reported by males compared to females, by younger compared to older employees, by smokers compared to non-smokers, and by clerical employees compared to nurses (Table 3).” 

We did not calculated a stepwise regression, but entered groups of  variables into the model  successively to be able to observe the amount of variance of previously included variables explained by variables included at a later step. This approach allows the reader to fully appreciate the effect of individual TPB variables, especially of rest break intention and the interactive term. To clarify this approach, we amended section 2.3 by adding: “The association between independent and dependent variables was analyzed using hierarchical linear regression analyses.” and “Thereby, separate models were calculated, i.e. groups of variables and/or individual variables were entered in the analysis successively, to be able to observe the amount of variance of previously included variables explained by variables included at a later step.”

Comment 8

It would also be a good idea to write the names of the dependent variables in the header of Table 3 as they appear in the table, because otherwise it will lead to confusion at first. It is indicated in the header rest break frequency, while in the table it appears Total rest breaks / hour. Although it is explained in the text, the table must be completely clear. You can write the header by including in parentheses the way that rest break frequency is described in the table.

We amended Table 3 accordingly.

Comment 9

On the other hand, the authors should clarify the meaning of the variable occupation*intention. It seems like an interaction that is not described anywhere, and that is not mentioned in the discussion either.

As stated above (Comment 6), we added information on the interactive variable in in section 2.2. and section 2.3. The interaction is also illustrated in Figure 1. To clarify that the figure illustrates the interaction, we amended the figure title: ”Effect of the interaction between occupation and rest break intention on rest break frequency”.  The corresponding hypothesis is stated in the last sentence of the introduction and is in line with TPB-reasoning.

Comment 10

Likewise, the profile of the type of person who seems to have a higher prediction of rest break frequency is not mentioned: it would be a male, younger than older, smoker, clerical employee with high behavioral control, rest break intention and occupation*intention.

The profile of the type of person having a higher prediction of rest break frequency is described in the results section, last paragraph: “Apart from the TPB variables, rest break frequency was also strongly associated with the individuals’ sex, age, smoking status and occupation. A higher rest break frequency was reported by males compared to females, by younger compared to older employees, by smokers compared to non-smokers, and by clerical employees compared to nurses (Table 3).”

Comment 11

And again, we still don't know the meaning of that interaction. I suspect that, since intention is measured with a Likert-type scale item and the reference group in occupation is clerical, this interaction indicates that clerical employees with the highest intention are those who predict the most rest break frequency. But this information has to be explained and the variable clearly described, as well as its meaning in the interaction, by the authors.

See responses to comment 9 as well as revised results section: “Finally, the association between rest break intention and rest break frequency was stronger for clerical employees (b = .05, t = 4.47, p < .001) than for nurses (b = .01, t = 2.31, p = .022), as can be seen in the significant interactive term (Table 3, Model 4), thus supporting Hypothesis 4. This relationship is illustrated in Figure 1 (slope tests and visualization of interaction effect with Excel template from Jeremy Dawson; http://www.jeremydawson.co.uk/slopes.htm). Rest break intention affected behavior to a greater degree in clerical employees than in nurses.”

Reviewer 2 Report

This well-presented and theory-based study explores the determinants of rest break behaviors of nurses and clerical staff in a German hospital. The study finds, as expected, that the theory of planned behavior is a relatively weak approach because it does not consider structural and contextual determinants of behavioral intentions. Individual level attitudinal and perception predictors of behavioral intentions explained less variance than did smoking  status, occupation (nurse vs clerical), sex and age. I think the authors could 1) say a bit more about the policies governing break behavior for each occupation in the study-- is there reason to expect that break behavioral intentions would not be influenced; 2) perhaps the most interesting finding was the lack of relationships between the social norms variables and behavioral intentions....these items had difficult grammar (in English, but perhaps not in German) and as the authors note reference what colleagues might think/feel about taking a break....it would be helpful in the methods or introduction section to speak more to measurement of subjective norms and behavioral control...and 3) to discuss any piloting of the questionnaire to understand how these questions were understood. 4) I think my biggest question about the paper lies with the discussion and conclusions: specifically, it would be valuable if the authors could sketch out what a revised TPB that better incorporates context might entail. and to further identify any practical use of the findings. I would also be interested in more discussion of the implications of the smoking findings....if breaks are so helpful, even necessary to good performance, and smokers and persons in less anxiety-provoking roles take more breaks, what should an employer do?

Author Response

Reviewer 2

Comment 1

This well-presented and theory-based study explores the determinants of rest break behaviors of nurses and clerical staff in a German hospital. The study finds, as expected, that the theory of planned behavior is a relatively weak approach because it does not consider structural and contextual determinants of behavioral intentions. Individual level attitudinal and perception predictors of behavioral intentions explained less variance than did smoking  status, occupation (nurse vs clerical), sex and age.

Thank you very much for this positive feedback.

Comment 2

I think the authors could 1) say a bit more about the policies governing break behavior for each occupation in the study-- is there reason to expect that break behavioral intentions would not be influenced.

We added some information on policies governing breaks in the introduction (one-before-last paragraph): “According to the Austrian Working Times Act (which applies to our sample), rest breaks of 30 or 45 minutes in total are mandatory for both occupational groups on days with more than six or nine hours in total. In addition, supplemental short rest breaks (< 15 minutes) are typical for both occupational settings [1, 9, 30].

Comment 3

2) perhaps the most interesting finding was the lack of relationships between the social norms variables and behavioral intentions....these items had difficult grammar (in English, but perhaps not in German) and as the authors note reference what colleagues might think/feel about taking a break ....it would be helpful in the methods or introduction section to speak more to measurement of subjective norms and behavioral control.

We slightly rephrased the items of the TPB-variables to be more in accordance with the German wording (Table 2).

We agree that the lack of relationships between the social norm variables and behavioral intentions is surprising. We discuss this lack of relationship in the discussion section (3rd paragraph), pointing out that one aspect of subjective norm (the descriptive norm) was omitted due to its conceptual closeness to behavioral control. We believe to have adequately presented theory and measurement of the TPB-variables in the introduction, including subjective norm and behavioral control. However, following the suggestion of the reviewer, we expanded our discussion of the subjective norm (discussion, 3rd paragraph): “However, keeping in mind the close association between the descriptive norm and behavioral control, the strong association of behavioral control with intention as well as rest break behavior as discussed below suggests that in contrast to the injunctive norm the descriptive facet of the subjective norm (the rest break behavior of colleagues) may indeed affect break behavior. Future studies will have to verify this explicitly.”

Comment 4

..and 3) to discuss any piloting of the questionnaire to understand how these questions were understood.

As pointed out in section 2.2. (Study Variables) “items were generated and phrased in accordance to the description of questionnaire construction stated in [33]. Additional items assessing salient beliefs were generated on the basis of nine interviews with nurses, clerical employees and technicians. In these interviews, advantages and hindrances of regularly taking rest breaks were explored.” Although we did not explicitly pilot the questionnaire, the statements of those interviewed resembled the generated TPB questions, suggesting that they mirror individual attitudes towards rest breaks.

Comment 5

4) I think my biggest question about the paper lies with the discussion and conclusions: specifically, it would be valuable if the authors could sketch out what a revised TPB that better incorporates context might entail and to further identify any practical use of the findings.

Thank you for this suggestion. We now stated suggestions for a more comprehensive model predicting rest breaks (discussion, last paragraph): “In a recent publication, Ajzen noted the potential of enriching the TPB by including background factors and variables shaping habit formation (43). Moreover, the Health Action Process Approach model from Schwarzer (44) tries to overcome the limitations of the TBP by describing the volitional processes of behavior formation (i.e., planning, ini-tiation, maintenance, and relapse management) in more detail and by adding additional factors such as situational barriers and resources, different types of self-efficacy, and self-monitoring. Beyond this social and health psychology perspective, however, such factors have not yet been systematically studied in the rest break literature (30). The integration of these variables into theoretical models would therefore be desirable in order to stimulate further research. For instance, a more comprehensive model predicting rest breaks should include situational factors with nudging properties such as task completion or a friendly interruption by a co-worker (45), work and organizational factors relating to impaired (e.g., time pressure) or successful (e.g., social support) recovery behavior, and individual motives for work breaks not primarily associated with recovery such as smoking or chatting (30, 46).”

Comment 6

I would also be interested in more discussion of the implications of the smoking findings....if breaks are so helpful, even necessary to good performance, and smokers and persons in less anxiety-provoking roles take more breaks, what should an employer do?

Thank you very much for this interesting comment. Since smoking status was a control variable in our study and we did not included a hypothesis in our paper, we decided not to expand on this topic in the discussion section. However, we did add the following statement (discussion section, one before last paragraph): “Indeed, the present study confirmed previous results that smoking employees take more breaks than their non-smoking counterparts {Wendsche, 2017 #4821}{Sarna, 2009 #3362}. Undoubtedly, smokers are more inclined to take breaks due to their addiction {Potvin, 2015 #4856}, but also social factors may play a role, such as escaping everyday routine {Murray, 1988 #5867}. Although traditionally smoking has been considered an eligible reason for work breaks, it would be desirable for employees to allow frequent breaks for all employees and to motivate and support nonsmokers in taking such additional breaks.”

Comment 7

English language and style are fine/minor spell check required

We conducted a spell and grammar check.

Round 2

Reviewer 1 Report

I think the authors have done a great job, especially including the mediating regressions with PROCESS. It seems to me that the article has improved a lot. Congratulations!

Author Response

Manuscript ID: healthcare-1296422 - Minor Revisions

Responses to reviewers, 2nd revision

We thank the reviewers for the renewed scrutiny of our manuscript and the elucidating suggestions made.

Reviewer 1:

  1. Line 52: We changed the sentence accordingly (“that rest breaks can also be effective..”).
  2. Line 58: Thank you for pointing this out. To make a stronger connection to the TPB, we extended the sentence as follows: “for example by changing employees’ attitudes toward breaks through information on benefits of rest breaks or by providing more control over work scheduling.”
  3. Line 118: To improve the argument why we chose these two samples, we rephrased the sentence to: “We chose to study these associations in a combined sample of nurses and clerical employees as these two occupational groups differ with respect to several key job characteristics such as occupational stress, job control and shift length. Statistically, this should increase the variance of the TPB variables and thus the reliability of the model while at the same time increasing the generalizability of the results.”
  4. Line 125: Yes, this is the correct title. We added the German original to clarify this point, thereby slightly modifying the sentence: “According to the applicable Austrian Working Times Act (”Arbeitszeitgesetz”)…”
  5. Line 137: To highlight the ramifications/aims of the present study, we added the following sentence: “The aim of the present study is to highlight relevant individual factors affecting employees’ rest break behavior. This should help devise evidenced-based strategies promoting the taking of rest-breaks which could be beneficial both for employees and for those administrating programs aimed at reducing occupational stress.”
  6. Line 141: We added more information to the recruitment procedure, now stating: “Administrative staff of the General Hospital of Vienna as well as nurses of various clinics of this hospital were contacted in February 2016 via e-mail by the heads of the corresponding departments (head nurse, head of the human resource department), who forwarded an invitation letter by one co-author (Theresa Tschulik) asking them to participate in an online-survey (www.soscisurvey.de) on rest-breaks during work-time. A link to the online survey was provided together with a password to access the survey.”
  7. Line 145: To clarify this point, we rephrased the sentence slightly to: “… as employees are required to take breaks if they work 6 hours or more per day, according to the Austrian Working Times Act.”
  8. Line 152: As the group of n=66 men are statistically meaningful, the gender ratio in both occupational groups was roughly equal and sex differences were controlled for statistically, we do not consider this a sever limitation of the study. Thus we did not follow the suggestion of the reviewer to add this to the limitation section of the paper.
  9. Line 182: To clarify this point, we rephrased the sentence: “As it is likely that nurses have less job control compared to administrative staff, e.g. less control over the timing of breaks due to patient care obligations (30) …”
  10. Line 210: We hope to have satisfactorily addressed this point in the introduction, line 118ff (see also point 3).
  11. Line 228: We modified the reference which is now contained in the references list as book.
  12. Line 242: We modified the reference which is now contained in the references list as journal article.
  13. Lines 258ff: We thank you for pointing this out and rephrased and shortened this paragraph (first paragraph of discussion).
  14. Line 275: The interpretation of this result (the limited impact of attitudes) is presented in this paragraph. To clarify this to a greater degree, we amended two sentences as follows: “Attitudes generally can be considered a relevant predictor of the intention…” and “There are two main explanations for the limited impact of intention in the present study.”
  15. Line 288: This is a good point. We therefore amended this section to illustrate that all employees may have insufficient knowledge of the benefits of rest-breaks. “Second, employees’ knowledge of health-related benefits of rest breaks generally may be insufficient, although differences between the two occupations were apparent. Nurses had a somewhat stronger positive attitude towards rest breaks than administrative employees, which might be a consequence of their greater health-literacy”.
  16. Line 309: Thank you for this suggestion. We added an explanation for the failure of the injunctive norm to affect intention or behavior and thus to support our hypothesis: “An explanation for the failure of the injunctive norm to affect intention as well as behavior may be that the taking of rest breaks as a means to promote well-being and performance are hardly an issue at the workplace, so that even if employees believed supervisors or colleagues wanted them to take breaks, they still may be wary about actually taking them. Thus, future studies would have to look into the actual discourse regarding rest breaks at the workplace.
    In addition, we modified our discussion on the descriptive norm to avoid possible misunderstandings and placed this in a separate paragraph to improve readability: “Regarding the descriptive norm (which effects we did not analyze as stated above), it should be pointed out that it presumably would have affected intention and behavior via behavioral control as discussed below, keeping in mind its close association with control. We therefore may speculate that an employee would be inclined to take breaks when colleagues do. Future studies would have to verify this assumption.”
  17. Line 314: We added a reference supporting our conclusion regarding control: “… which is in line with a recent study on factors affecting rest break behavior in nurses in Germany [41].”
  18. Line 337: To specify, we added: “… for example by interruptions or time pressure,”
  19. Line 375: Thank you. We added a reference to this paragraph “Furthermore, our research focused on two specific occupations differing in control over working time. It remains unclear to what extent our results can be generalized to other occupations with different job characteristics because lower time control is positively related to workload [50], which might further impair break behavior [30, 41]. This should be investigated in future research additionally by assessing key job characteristics.
  20. Line 387: Thank you for this recommendation. We added some practical implications “Our results suggest that organizations should increase employees’ temporal flexibility (i.e., more control regarding the duration and timing of tasks during the workday) if they intend to improve their break frequency. The limited impact of intention also suggests that making use of situational factors in terms of nudging (i.e., digital break reminders or ‘break’ posters on the walls of the work site) may help to foster rest-taking behavior.“

Reviewer 2:

We edited the references and used square brackets.